

# Marine heatwave amplifies extreme multi-hazards of extratropical cyclone Babet

Piyali Goswami[1], Ségolène Berthou[2], Theodore G. Shepherd[1], Ambrogio Volonté[1,3], Sana Mahmood[2], Juan Manuel Castillo[2], Anne-Christine Péquignet[2], Yong-June Park[2], Mark Worsfold[2], Regina Rodrigues[4], and Magdalena A. Balmaseda[5]

[1]Department of Meteorology, University of Reading, Reading, UK
[2]Met Office, Exeter, UK
[3]National Centre for Atmospheric Science, Reading, UK
[4]Department of Oceanography, Federal University of Santa Catarina (UFSC), Brazil
[5]European Centre for Medium-Range Weather Forecasts (ECMWF), Reading, UK

**Correspondence:** Piyali Goswami (p.goswami@reading.ac.uk)

**Abstract.** Climate change is increasing marine heatwave (MHW) frequency and intensity and intensifying rainfall from extratropical cyclones. Storm Babet, the United Kingdom's most impactful 2023 extratropical cyclone, followed ~7 weeks of MHW conditions in the North Sea and cooled it by ~1°C. The storm produced extreme rainfall, widespread flooding, and severe coastal erosion across northern regions of the UK. Its prolonged southeasterly winds generated above 8 m significant
wave heights and up to 60 cm surge in the North Sea coastline of the UK. We show the MHW was generated at the surface by persistent anticyclonic conditions: clearer skies boosted shortwave heating, weaker winds reduced mixing and latent cooling. Subsequent short spells of cyclonic conditions allowed heat to penetrate at depth and be retained throughout the depth of the shallow North Sea. Using a state-of-the-art regional coupled model, we assess how pre-existing marine heatwave conditions influenced the storm's evolution and associated hazards. The warm shelf seas boosted evaporation, energised the boundary
layer and deepened the cyclone. Indeed, the fast, relatively dry and cold southeasterly low-level jet saw amplified latent and sensible heat fluxes over the MHW. These extremely intense air-sea fluxes led to the decay of the MHW whilst loading the storm with additional moisture. Relative to a cooler ocean counterfactual, the MHW amplified multihazard outcomes across the Northeast of the UK, increasing river discharge (12-18%), coastal wave power (~9%), and storm surge (~20%). These results highlight the role of MHWs as amplifiers of multi-hazard storm impacts over the shallow Northwest European Shelf,
where extra ocean heat cannot escape to depth.

**Keywords.** Marine heatwaves, North Sea, coastal multi-hazard, extratropical cyclone, regional coupled models.

## 1 Introduction

Marine heatwaves (MHWs) are periods of anomalously high sea surface temperature (SST) lasting from days to months (Hobday et al., 2016). Their frequency, duration, and intensity have increased significantly over recent decades due to anthropogenic
climate change (Frölicher et al., 2018). These events have major consequences for marine ecosystems and fisheries, often triggering habitat loss and species redistribution (Smale et al., 2019; Smith et al., 2023) and sometimes co-occurring with other



oceanic extremes such as high acidity and low chlorophyll (Rodrigues et al., 2025). Beyond their ecological effects, MHWs influence the atmosphere by enhancing surface heat and moisture fluxes (Berthou et al., 2024), modifying boundary-layer stability, and altering the development and intensity of weather systems (Choi et al., 2024). While the impacts of MHWs on tropical cyclones and Mediterranean cyclones are increasingly documented (Frölicher et al., 2018; Dzwonkowski et al., 2020; Jangir et al., 2024; Choi et al., 2024), their role in fueling extratropical cyclones, especially at the end of the North Atlantic extratropical track region, remains poorly understood.

The northwest European shelf (NWES) is a shallow continental shelf (depth $< 250$ m) and has experienced recurrent MHWs in recent years (Chen and Staneva, 2024), raising concerns about their influence for marine ecosystems and regional climate hazards. Observations show that SST around the UK has increased by approximately 0.3°C per decade since the early 1980s, with the strongest trends exceeding 0.4°C per decade in the southern North Sea (Cornes et al., 2023a). The frequency of MHWs around the British Isles has also increased by about four additional events per year relative to the late twentieth century, with the highest rise in northern waters. Projections under a high-emission scenario (RCP8.5) indicate that mean SST over the North Sea could be about 3°C warmer by the end of the century (Cornes et al., 2023a). Recent studies documented the unprecedented marine heatwave of June 2023 over the NWES, linked to persistent anticyclonic conditions and strong solar heating (Berthou et al., 2024; Atkins et al., 2025). Such events highlight the potential for shallow shelf seas to accumulate heat and enhances the potential for MHWs to interact with overlying storms and modify regional climate extremes.

Extratropical cyclones are the primary drivers of winter precipitation, wind, and coastal flooding in northwest Europe (Hawcroft et al., 2018). Rapid attribution studies show that winter storms affecting northwest Europe during the 2023/2024 season produced more precipitation per event due to anthropogenic warming (Kew et al., 2024), consistent with projections from both global and regional climate models. Recent modelling studies indicate that precipitation associated with extratropical cyclones will intensify in a warmer climate, even if the total number of storms declines. Both global and regional simulations show that rainfall in intense cyclones increases with surface temperature at rates close to the Clausius-Clapeyron relation (about 7 % K$^{-1}$), driven mainly by enhanced atmospheric moisture rather than stronger storm dynamics (Hawcroft et al., 2018; Kodama et al., 2019). Convection-permitting models further suggest an increase in convective activity and heavier rainfall within extratropical cyclones over northern Europe (Berthou et al., 2022).

Despite these advances, most studies have focused on the atmospheric component of storm intensification, while the role of oceanic anomalies, particularly MHWs, remains poorly constrained. The NWES, with its shallow bathymetry and rapid ocean-atmosphere feedbacks, provides a natural testbed for exploring this coupling. Warm sea anomalies can modify surface fluxes and boundary-layer structure, potentially influencing cyclone development, intensity, and precipitation. Understanding these interactions is critical for assessing how current MHWs, representative of future average warming levels, may amplify compound hazards such as extreme rainfall, storm surge, and coastal flooding.

In October 2023, an exceptional marine heatwave persisted over the North Sea for nearly seven weeks before the arrival of Storm Babet, a high-impact extratropical cyclone that caused widespread flooding, coastal erosion, and infrastructure damage across the UK. Storm Babet was among the most intense autumn storms on record for the region, providing a clear case to examine ocean-atmosphere coupling under extreme conditions. This co-occurrence offers an opportunity to quantify the





influence of a pre-existing marine heatwave on storm evolution and its associated hazards using a storyline approach (Shepherd, 2016). In this study, we use the Regional Coupled System UKC4 (Berthou et al., 2025a) in two modes: first as a constrained regional coupled hindcast system to identify the causal factors leading to the development of the autumn 2023 MHW over the

northwest European shelf; and second, as a 5-day regional forecast system run under factual and counterfactual conditions to quantify how the MHW affected the multi-hazard characteristics of Storm Babet. Section 2 describes the model configuration, experimental design, and methods used followed by an analysis of the marine heatwave and Storm Babet evolution in Section 3. Section 4 quantifies the influence of the MHW on storm intensity and associated hazards, and Section 5 discusses the broader implications for compound climate risks.

## 2  Methods

### 2.1  Observations

We used in-situ observations from river gauges in Scotland from the Scottish Environment Protection Agency, significant wave height and wind data from the WAVENET network, oil rigs and buoys maintained by other European countries. Surge data came from the UK National Tide Gauge Network owned and maintained by the Environment Agency. We used the SST data

from Operational Sea Surface Temperature and Ice Analysis (OSTIA; Good et al., 2020) for the calculation of MHW and National Climate Information Centre (NCIC; Hollis et al., 2019) daily gridded data for precipitation.

### 2.2  Backward trajectories in deterministic global forecasts

To determine the origin of air parcels, we use global operational Met Office forecasts starting on the day before the air parcels are started from. The use of the global model enables to use the Lagrangian Analysis Tool (LAGRANTO, Wernli

and Davies,1997; Sprenger and Wernli, 2015), to compute forward and backward parcel trajectories from gridded winds. LA-GRANTO integrates parcel positions with a three-time iterative forward-Euler scheme and an internal step equal to 1/12 of the input interval; we use 3-hourly inputs (15 min interval), which are standard for synoptic-scale trajectory analyses (Volonté et al., 2020; Rai and Raveh-Rubin, 2023; Deoras et al., 2024). Winds and scalars are linearly interpolated in space and time between forecast fields. The model outputs parcel positions and interpolated diagnostics (e.g. pressure, temperature, specific

humidity, potential temperature) at a user-defined (hourly in our case) sampling interval.

### 2.3  Met Office Regional Coupled System: UKC4

This study utilises the fourth generation of km-scale regional atmosphere-ocean-wave coupled system, as documented in a parallel paper (Berthou et al., 2025a), originally described by Lewis et al. (2019). The domain is centered on the UK and covers all the northwest European continental shelf seas, spanning approximately 46°N-63°N and 19°W-13°E. It operates on

a rotated pole coordinate system, with the pole origin positioned at 177.5° longitude and 37.5° latitude. The system is fully coupled, facilitating two-way feedback between its components, which include the Met Office Unified Model (UM) for the



atmosphere (Brown et al., 2012), the Nucleus for European Modelling of the Ocean (NEMO) for the ocean (Madec and the NEMO team, 2016), and WAVEWATCH III®for the wave component (Tolman and the WWIII development group, 2014).

### 2.3.1 UM Regional Atmosphere and JULES Land Component

The atmospheric component of UKC4 (the Unified Model, version 13.5) is implicitly coupled with the Joint UK Land Environment Simulator (JULES) for land surface and additional river routing is provided by the River Flow Model, as described in Lewis and Dadson (2021). The regional atmosphere and land configuration RAL2 is used, documented in Bush et al. (2023). This configuration was chosen as it was operational for weather forecasts issued by the Met Office for storm Babet.

The domain is a rotated pole curvilinear grid with variable-resolution, with horizontal grid lengths ranging from 2.2 km in
the central domain to 4.4 km in the outer domain, named ENUK domain (Porson et al., 2020). In the vertical, the domain is distributed in 70 vertical levels, with model top around 40 km. These resolutions enable an explicit representation of convective processes: deep and shallow convection are explicitly resolved by the model dynamical core (Bush et al., 2023). A timestep of 60 s is used. JULES land model generates runoff in two ways: i) infiltration excess runoff, which is mainly a function of rainfall intensity and saturation fraction of the first 1m of soil (generated by the Probability Distribution Model) and ii) sub-surface
runoff, generated by Darcy flux at the bottom of the 3m soil column (Best et al., 2011). Surface and subsurface runoffs are routed with separate speeds by the RFM model. Both the PDM and RFM models have constant parameters over the whole UK.

### 2.3.2 NEMO Regional Oceanic Component

The ocean component uses the Atlantic Margin Model 1.5 km domain (AMM15) with the Coastal Ocean 8 (CO8) configuration (Graham et al., 2018; Tonani et al., 2019) and employs NEMO version 4.0.4 on a 1.5 km resolution on a curvilinear grid with
the same rotated pole as the atmosphere. The vertical discretisation consists of 51 hybrid z-s levels, which means fixed depth levels in the deeper part of the domain, and terrain-following levels in the shallower part, with a transition zone on the shelf-break. The 1 m constant surface thickness guarantees uniform surface heat fluxes. The model incorporates a non-linear free surface formulation and an energy-conserving momentum advection scheme. Lateral boundary conditions are prescribed using a free-slip approach (Tonani et al., 2019).

Tidal forcing of Finite Element Solution-2014 (FES2014) is implemented using 11 tidal constituents (M2, S2, N2, K1, O1, Q1, M4, K2, P1, MS4, MN4), applied via a Flather radiation boundary condition (Flather, 1976), and further refined by incorporating an equilibrium tide representation. Heat and fluxes are provided directly by the regional atmospheric model. Mass, momentum, and energy exchanges between the atmosphere and the underlying land and ocean surfaces are modeled using JULES (Best et al., 2011). Heat fluxes are passed directly to the ocean, while 10 m neutral winds are passed to the wave
model, which calculates the momentum flux and sends it to the ocean. In this study, NEMO uses a baroclinic time step of 60 s, with the barotropic time step set to one-thirtieth of that value. The AMM15 model has two unstructured lateral boundaries to the north, west, and south with the Atlantic Ocean, and to the east with the Baltic Sea, provided by Copernicus Marine (CMEMS) models. The Atlantic boundary comes from global Marine Forecasting Centre (GLO-MFC) global ocean model PSY4 which is based on NEMO ORCA12, which supplies hourly Sea Surface Height, daily mean temperature and salinity





at all depths, and daily barotropic velocity. The Baltic boundary uses data from BAL-MFC, also based on NEMO, providing

hourly temperature, salinity, and barotropic velocity. Rivers use climatological values, as described in Tonani et al. (2019).

### 2.3.3 WAVEWATCH III Regional Wave Component

The wave component is based on WAVEWATCH III (WWIII) version 7.13, which has been modified to support coupling
exchanges (Castillo et al., 2022). The model domain covers the northwest European shelf, employing a two-tier Spherical
Multiple-Cell grid refinement. In the open ocean, grid resolution is approximately 3 km, whereas in coastal regions where the
water depth is less than 40 m, resolution increases to 1.5 km to better capture nearshore processes. The timestep for WWIII is
600 s.

Wave dynamics are parameterised using the ST4 package (Ardhuin et al., 2010), which accounts for wind-wave interactions,
whitecapping dissipation, and swell attenuation. To further refine wave behavior in shallow waters, the model includes the
surf-breaking parameterisation proposed by Battjes and Janssen (1978) and applies the JONSWAP bottom friction formulation
(Hasselmann et al., 1973) to represent wave energy dissipation due to seabed interactions. These parameterisations ensure that
the UKC4-Wave configuration is well-suited for intermediate and shallow water environments, such as those characteristic of
the northwest European shelf.

### 2.3.4 Coupling Framework

Coupling between these components is achieved through the Ocean-Atmosphere-Sea Ice-Soil (OASIS-MCT) coupler (Valcke,
2013). The atmosphere sends 10 m neutral winds to the wave model and pressure and heat fluxes to the ocean model. The
waves then send the surface stress, significant wave height, mean period and Stokes drift to the ocean. The ocean feeds back
the temperature and currents to the atmosphere, and the currents and total water depth to the waves. The waves send the
Charnock parameter back to the atmosphere. The model is configured with a 1 h coupling frequency. Note the rivers are not
fed into the ocean in this particular case; this capability is currently under development.

### 2.4 Forecast Set Ups for UKC4

In this article, we use UKC4 in two modes: i) 2 month hindcasts started at 00UTC 21 August 2023 and 21 August 2020, with
atmosphere and ocean Lateral Boundaries (LBCs) reset to day 1 global forecasts every day to diagnose the drivers of the 2023
North Sea MHW by following the observed synoptic evolution and contrasting with a near climatological year (2020); ii) 5-day
ensemble forecasts started 00UTC 17 October 2023, with atmospheric LBCs coming from MOGREPS-G global forecasts, and
deterministic ocean LBCs from GLO-MFC and BAL-MFC for the Atlantic Ocean and Baltic Sea boundaries, to quantify how
the pre-existing MHW modified Storm Babet and its multi-hazards under identical large-scale forcing via a factual (ocean
initialised 17 Oct 2023) and counterfactual (ocean initialised 17 Oct 2020) pair.

The ensemble setup consists of the 18-member Ensemble-Regional Coupled Suite framework, which integrates the Met
Office Global and Regional Ensemble Prediction System over the United Kingdom (MOGREPS-UK, Hagelin et al., 2017;



Porson et al., 2020) atmosphere-land ensemble forecast with the regional ocean-wave system. It includes one unperturbed reference simulation and 17 perturbed members, where atmospheric initial and lateral boundary conditions are generated by downscaling perturbations from the global MOGREPS-G system. While the atmospheric component is ensemble-based, the ocean and wave components are deterministic, with coupling handled consistently across all members. SST perturbations are

applied from MOGREPS-G SST perturbations through the OASIS coupler, and only seen by the atmospheric system. They are kept fixed for the whole forecast and they are applied so that their average across all ensemble members is 0 and their maximum amplitude is 2°C (Tennant and Beare, 2014). These member-indexed SST perturbations are used operationally to correct under-dispersion of near-surface air temperature; they act as spread inflation rather than representing observational SST uncertainty. They are fixed in time and zero-mean across members so that reliability of 2-m temperature is improved without

biasing the ensemble mean. It was chosen to keep these perturbation in the coupled system to stay closest to the operational set-up. Further technical detail on the ensemble set up is available in Gentile et al. (2022).

Two ensemble 5-day forecast simulations were triggered for storm Babet, both starting on 17 October 2023. The ocean model is either initialised from 17 October 2023, 2 days before the storm starts in the MHW experiment, or 17 October 2020, 3 years and 2 days before the storm in the no_MHW experiment. 2020 is chosen as the near-climatological baseline since its SST

is closest to the long-term and 2003-2022 climatology across the North Sea region, providing a representative, dynamically consistent ocean state for the no_MHW run (Figure 1a), and is a year for which an operational forecast is available (2018-now). The marine heatwave is slightly underestimated compared to the observed anomaly in the North Sea (Figure 1b,c). We can therefore consider the coupled experiment as a lower estimate of the marine heatwave signal on the storm.

Although SST perturbations are applied to these runs in order to stay as close as possible to the operational set-up, individual

members are perturbed in the exact same way between MHW and no_MHW ensembles, so that the SST difference seen by the atmosphere and shown in Fig. 1c is the same for each member pair, as the inter-ensemble differences reflect only the ocean initialisation.

We analyse two regions: the Central North Sea region (Black polygon outlined in Figure 1b) for the ocean response, and a fixed rectangular precipitation box in northeast Scotland (red rectangle in Figure 3). Hereafter we refer to this as the Eastern

Scotland box [4.0°W, 55.5°N, 2.0°W, 57.5°N].

## 2.5 Marine heatwave detection

Marine heatwaves were detected following Hobday et al. (2016). We used OSTIA daily SST with a fixed climatology over 1982-2011 to compute the seasonally varying 90th-percentile threshold using a ±11-day moving window. A MHW is defined as ≥5 consecutive days above this threshold, with gaps of ≤2 days merged. We use the Hobday et al. (2018) scheme to assign

categories which scales event intensity. For category maps (Figure 1b) we use a rolling 7-day majority (5-of-7) rule: for each grid cell, check the most recent seven days. If at least five of those days show a marine heatwave (category 1 or higher), we plot the cell as MHW and use the category from the most recent day; otherwise we plot category 0.





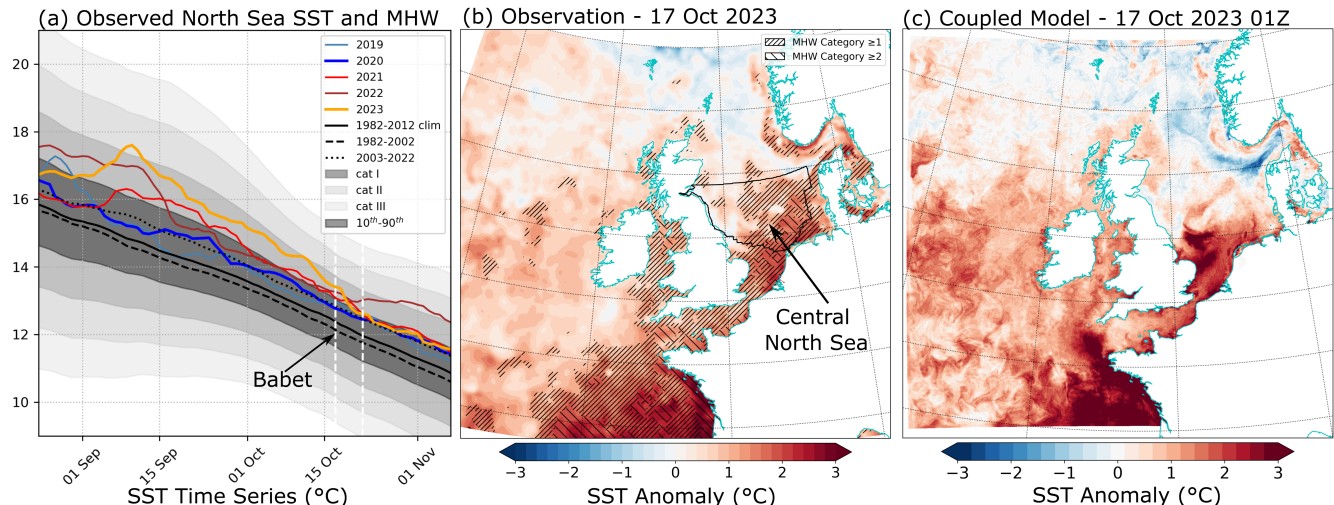

**Figure 1.** a) SST time series (°C) from the OSTIA dataset averaged over the North Sea. The solid black line is the long term climatology (1982-2012), dashed black line is the climatology of 1982-2002, dotted black line is recent climatology (2003-2022) which is more close to SST of 2020. The white dashed lines show the time window of storm Babet b) SST anomaly from 1982-2022 climatology from OSTIA, the black polygon represents central North Sea c) SST difference between the 17 October 2023 and 17 October 2020 ocean restarts used in the coupled run.

## 2.6 Mixed-layer heat budget

To quantify how atmospheric forcing translates into ocean warming, we compute a mixed-layer heat budget on hourly output
from the 2023 and 2020 hindcast regional coupled simulations over the central North Sea (Figure 2). This region was chosen
as a region of homogeneous hydrodynamic regime (Wakelin et al., 2012). For every grid cell in the box, we formed the budget
terms and then averaged them over the region. The mixed layer temperature tendency ($\frac{dT}{dt}$) can be expressed as a balance
between surface heat fluxes, shortwave radiation penetrating below the surface, entrainment at the base of the mixed layer, and
residual processes (advection, diffusion, and unresolved terms) as in Berthou et al. (2024). The governing equation is:

$$\frac{dT(t)}{dt} = \frac{Q_{\text{SW}}\left(1 - (1-0.58)\text{e}^{-\frac{h(t)}{\xi_{sw}}}\right)}{\rho C_{\text{p}}h(t)} + \frac{Q_{\text{LH}} + Q_{\text{SH}} + Q_{\text{LW}}}{\rho C_{\text{p}}h(t)} - \epsilon \frac{T(t-1) - T_{dh}(t-1)}{h(t)} \frac{dh(t)}{dt} + R$$

Here, the term on the left, $dT(t)/dt$, represents the temperature tendency of the surface mixed layer (SML) averaged over
hour $t$. The terms on the right-hand side represent the physical processes driving this change. The first two terms represent
the net surface heat flux normalized by the thermal inertia of the SML ($\rho C_p h$) where $\rho$ is the seawater density (1027 $kg/m^3$),
$C_p$ is the specific heat capacity of water (3850 $JKg^{-1}°C^{-1}$) and $h(t)$ is the SML depth (m). The first term accounts for
the depth-dependent absorption of penetrating shortwave ($Q_{\text{SW}}$) represented using an exponential decay with an e-folding
depth $\xi_{\text{sw}}$ (15 m) following Paulson and Simpson (1977), where the second term combines the non-penetrating surface fluxes:





latent heat ($Q_{\text{LH}}$), sensible heat ($Q_{\text{SH}}$), and longwave radiation ($Q_{\text{LW}}$). The third term quantifies the temperature change due to entrainment at the base of the mixed layer; this term is active only when the SML deepens ($dh(t)/dt > 0$), as governed by the Heaviside function, $\epsilon$. It depends on the temperature difference between the SML at the previous timestep, $T(t-1)$, and the

temperature of the water being entrained from below, $T_{\text{dh}}(t-1)$. $R$ is a residual term that includes all other processes, such as advection, diffusion and other non-resolved processes.

## 3    Marine heatwave state and storm Babet description

### 3.1    Drivers of the autumn 2023 Marine Heatwave

To identify the physical drivers responsible for the formation of the 2023 MHW, we contrast the atmospheric and oceanic

conditions of that year with the same period in 2020, a year that did not experience a MHW and serves as a near-climatological reference. The analysis is focused on the central North Sea region, which the storm air streams crossed before hitting Scotland.

The primary catalyst for the MHW was a sequence of persistent atmospheric anticyclonic weather regimes in September 2023. As shown in Figure 2a, the period leading up to and during the MHW's peak (20th August - 20th September) was dominated by regimes 1, 3, 6, 12, 16 which are all characterised by weak winds (Neal et al., 2016), and most with anticylconic

conditions over the North Sea. High-pressure systems (anticyclonic conditions) suppress cloud formation and reduce wind speeds, favouring marine heatwaves (Berthou et al., 2024). The prolonged presence of these regimes created an extended window of clear skies and calm seas, maximizing the amount of solar radiation reaching the ocean surface. Weather regimes between 1 and 10 correspond to weak circulation, more typical of summer, while regimes between 20 and 30 represent strong circulation, more typical in winter. Regimes 10-20 are intermediate, most frequent in autumn and spring. In contrast to 2023,

the atmospheric conditions in 2020 were characterised by variable weather regimes, with many occurrences of weather regimes above 10 and 20, with stronger winds (Figure 2d). The frequent shifts between different patterns resulted in more unsettled weather with greater cloud cover and higher wind speeds, which intermittently limited the solar energy input to the surface mixed layer of the ocean.

The heat budget analysis of the two periods shows that the 2023 MHW was driven by a strong and sustained net heat gain in

a shallow mixed layer. The event's evolution can be broken down into four distinct phases which are labeled P1-P4 in Figure 2.

The initial phase of rapid formation (P1), occurring from approximately late August to September 8th, was initiated by persistent high-pressure weather regimes (3,6,16) or a weak circulation weather regime (1). This delivered an intense accumulation of shortwave radiation (orange line), creating a potential warming of +7°C over the event's full duration. This forcing was critically amplified by a pre-conditioned shallow mixed layer depth (MLD) of only 5-20 meters (Figure 2c). The MLD

shown in panels (c) and (f); represents the upper ocean layer where temperature, salinity and density are relatively uniform due to turbulent mixing. It acts as the ocean's immediate heat reservoir, determining the volume of water over which the net surface heat gain or loss is distributed. This shallow layer concentrated the heat gain near the surface, and with minimal entrainment cooling during this period, the cumulative temperature tendency rose sharply (+1.8°C).



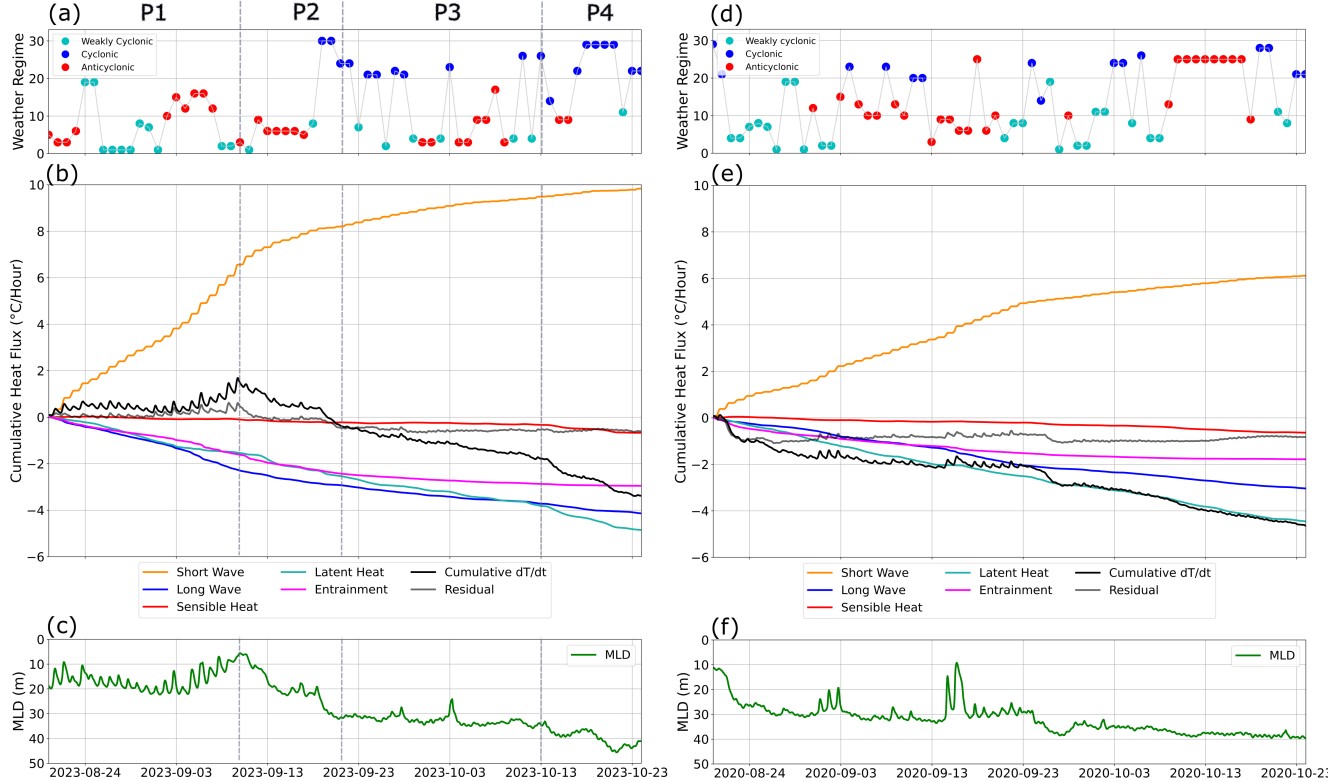

**Figure 2.** Atmospheric and oceanic drivers of 2023 marine heatwave (left column, a-c) contrasted with conditions during the same period in the non-MHW year of 2020 (right column, d-f). The top panels show the evolution of the synoptic weather regime using a 30 regime classification for Northwest Europe (Neal et al., 2016); coloured by regime category: Weak (cyan), Low (blue), and High (red) (a, d). The middle panels show the cumulative heat flux tendency components in °C/Hour (b, e), including Short Wave (orange), Long Wave (blue), Sensible Heat (red), Latent Heat (cyan), Entrainment (magenta), and a Residual (grey) term. The net cumulative temperature tendency (Cumulative dT/dt) is shown in black. The bottom panels show the mixed layer depth (MLD) in meters (c, f)

The second phase (P2), lasting until September 21st, marked a transition where the rate of warming decelerated, as the anomaly deepened. Although strong solar input continued, significant cooling fluxes began to more effectively counteract it. The most critical change was the onset of substantial entrainment cooling (magenta line) as the mixed layer deepened from 8m to 30m, which worked in concert with slow ongoing latent heat and long wave radiation losses. The mixed layer deepening was due to stormy conditions (weather regime 30) on September 19-20. This mixing phase culminated around September 21st, when the heat budget was momentarily balanced (Cumulative dT/dt = 0). This caused the reduction in intensity of the surface MHW in mid-September when the SST anomaly was approximately +1.5°C above the climatological mean (Figure 1a), but more importantly, it meant the excess-heat anomaly deepened, storing the heat in deeper layers of the North Sea, away from





cooling mechanisms at the surface. This behaviour is consistent with glider observations (Figure S1-2), supporting confidence in the model's vertical temperature structure.

Subsequently, the MHW transitioned into a prolonged phase of persistence (P3), lasting from late September through mid-
October. This stage was characterized by a slow but steady net heat loss, as indicated by the gentle downward slope of the cumulative temperature tendency (Figure 2b). This stability occurred as the deep mixed layer prevented entrainment cooling from below, while changes in weather pattern simultaneously suppressed the primary heating from solar radiation. With both major heating and cooling dynamics neutralized, the MHW's very slow decay was governed primarily by the persistent surface cooling from latent and longwave fluxes, allowing it to maintain a high intensity for nearly a month.

The prolonged period of persistence in P3 was brought to an abrupt end by storm Babet around 20 October. It was triggered by a rapid cooling dominated by surface heat fluxes. In the shallow (40-50 m) Central North Sea, the water column was already well-mixed in P2, making entrainment a negligible factor. Instead, the storm's intense winds drove large increase in latent heat flux, forcing a rapid loss of heat from the sea surface to the atmosphere. This efficient heat extraction quickly dissipated the remaining thermal anomaly throughout the water column, bringing the MHW to a conclusion.

The 2020 period provides a climatological example. The weaker shortwave radiation input provided a more modest potential warming of only +4.5°C for the same period. This energy gain was more than compensated for by the combined cooling effects of latent heat (-2.5°C), long wave radiation (-2°C), and entrainment (-1.5°C) resulting in a net cooling of approximately -2°C.

In summary, the 2023 MHW was the result of a synergistic combination of factors: persistent high-pressure weather regimes delivered an anomalous amount of solar radiation to the ocean, and anomalously weak winds favoured a shallow mixed layer,
which trapped this heat at the surface, amplifying the increased solar radiation effect. The absence of this atmospheric forcing and oceanic susceptibility in 2020 prevented MHW formation and instead led to a net cooling of the surface ocean, more typical of September and October in this region.

## 3.2 Storm Babet multi-hazards and performance of the coupled ensemble forecast

Storm Babet brought exceptional rainfall to parts of eastern Scotland with 150 to 200 mm falling in the wettest areas (Figure
3a) and the Met Office issued two red warnings for rain, as documented by Kendon (2023). Heavy, persistent and widespread rain also affected much of England, Wales and Northern Ireland from 18-20 October, with 100 mm falling fairly widely across the UK (Figure 3a). Storm Babet also brought some very strong winds, with gusts over 50 Knots (58 mph) across north-east England and much of Scotland. A blocking area of high pressure over Scandinavia prevented the two low pressure systems responsible for storm Babet from progressing eastwards into Scandinavia. This resulted in sustained wind speeds over the
North Sea lasting over 48 h. Storm Babet brought a combination of in-land and coastal damage; the river Tay and South Esk burst their banks; beaches in Musselburgh saw the equivalent of a 5-year erosion rate (3 m retreat in Montrose, near the mouth of river South Esk); harbour walls in North Berwick collapsed in sections; a North Sea drilling platform lost anchors; and the storm resulted in 7 deaths. Thankfully, the storm hit during a week of neap tide, but its surge lasted for several tidal cycles.

Figure 3 shows that the ensemble forecast was of good quality for the eastern part of Scotland for 5-day accumulated
precipitation, the region where the red warning was in place. As this is a single case, formal quantitative verification is not




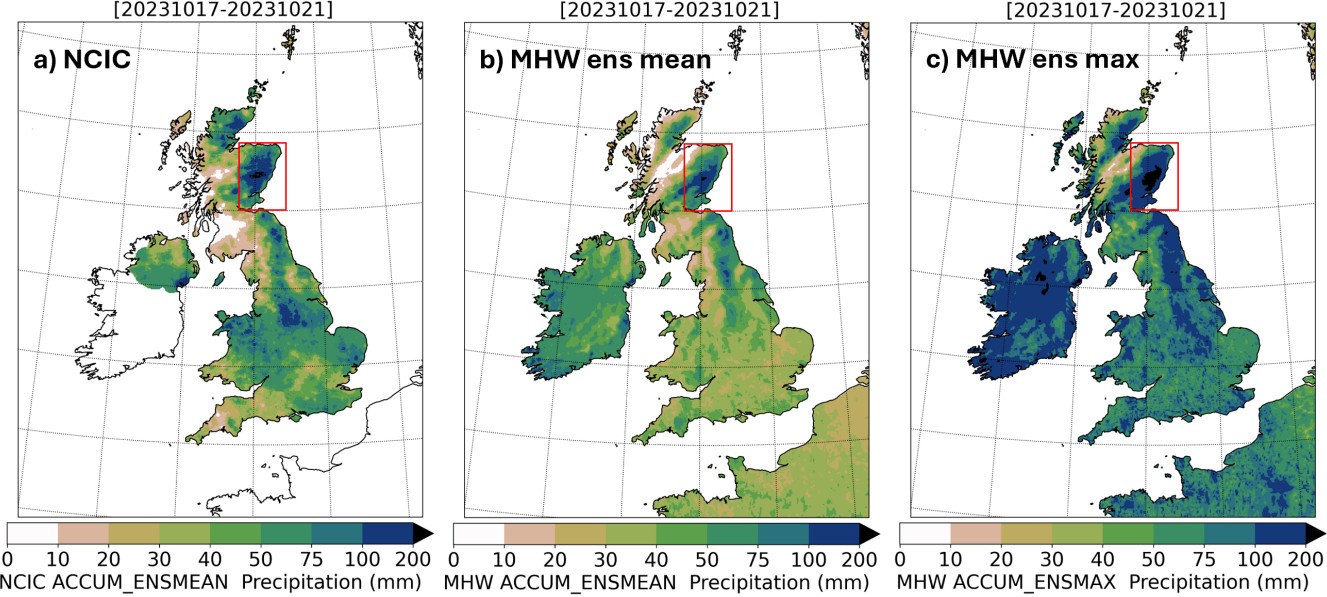

**Figure 3.** Accumulated precipitation (mm) during 17-21 October 2023: (a) NCIC daily gridded observations, (b) 5-day coupled forecasts (ensemble mean), and (c) ensemble maximum across members. The red box marks Eastern Scotland, the focus area with highest precipitation.

meaningful, so we restrict ourselves to a qualitative assessment. In this region the forecast behaved near-deterministically. Elsewhere, higher observed totals fall within the ensemble envelope (seen in Figure 3c), indicating the system can produce such amounts, but those regions are outside our scope. We therefore focus on the Eastern Scotland box in Figure 3, the primary impact area with concurrent inland and coastal hazards; the forecast there is sufficiently accurate for the process analysis that

follows.

Figure 4 presents a comparison between observations (black) and the coupled ensemble forecast (orange) for multiple hazards associated with Storm Babet. The panels show river discharge for the South Esk, surge and tide at Leith tide gauge, river discharge for the Tay, and significant wave height at the Firth of Forth WaveNet buoy. In both the South Esk and Tay catchments (panels a and c), observed river discharge shows a rapid increase during the storm, consistent with the heavy rainfall. While the

ensemble forecast reproduces the overall rising trend in river discharge, it lags 18 h behind the observations in capturing the peak. For each river, a few members capture the correct amplitude of river discharge increase generated by the storm, which shows good skill two days before the storm starts. Upstream observations show better model/observation agreement, which suggests the river speed is too slow in the coupled system (not shown).

At Leith (panel b), the tide gauge provides measurements of total water level, which can be decomposed into tidal and surge

components. Although the events occurrence during neap tide limited the overall hazard compared to a spring tide scenario, both high and low tides were higher relative to climatology by 0.3-0.6 m, reflecting the influence of storm surge. This storm surge was likely caused by the combined effect of low atmospheric pressure, which raises sea level, and strong winds that drove





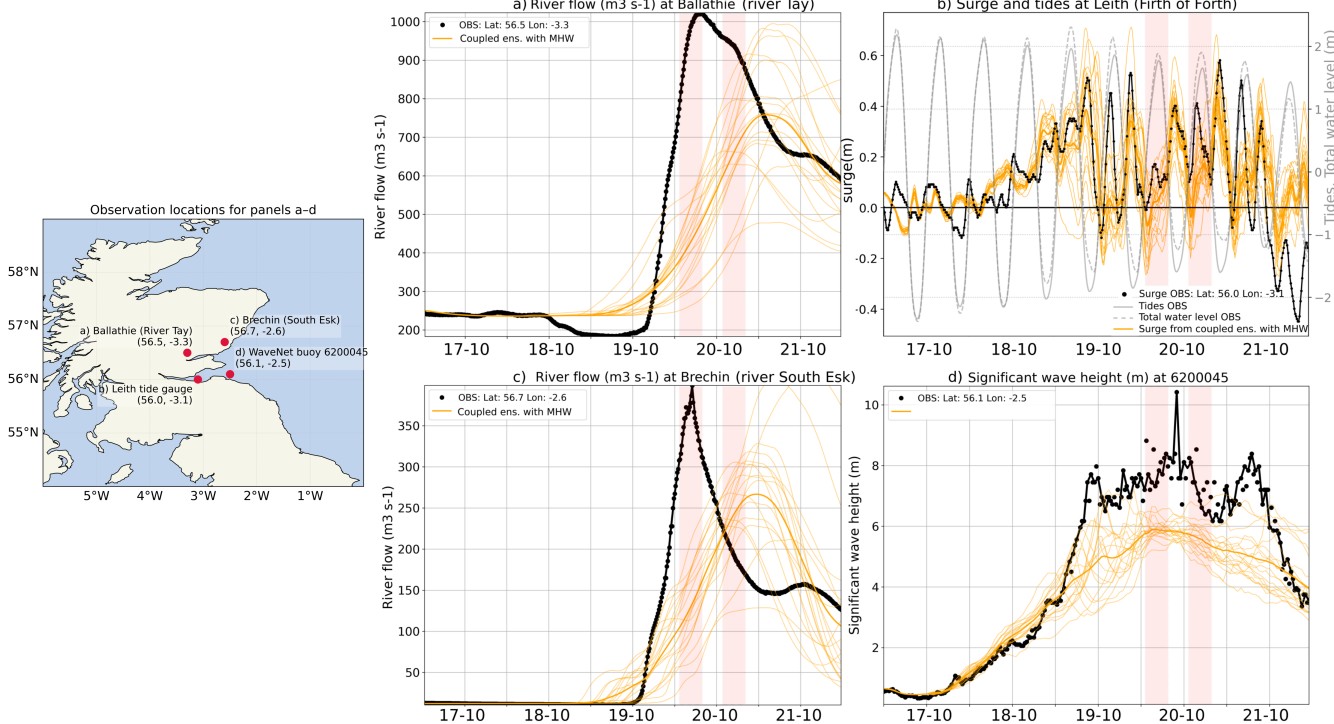

**Figure 4.** Multi-hazard time series for Storm Babet comparing observations (black) with the coupled ensemble forecast including MHW (orange; thin = members, thick = ensemble mean). Panels: (a) river flow ($m^3 s^{-1}$) at River Tay (closest available gauge to sea); (b) Leith tide-gauge water level (m) showing observed total (black), astronomical tide (grey), and modelled surge (orange); (c) river flow ($m^3 s^{-1}$) at River South Esk (closest available gauge to sea); (d) significant wave height (m) at the Firth of Forth WaveNet buoy. In panels a-c, model-observation mean of the first day of the run was removed from the model data to remove the model bias in pre-storm conditions and account for different reference levels (sea surface height). Red vertical bands mark intervals when hazards coincide.

water toward the coast. Overall, some ensemble members of the coupled ensemble capture the surge amplitude of 0.3-0.6 m, although the ensemble mean shows a 0.1-0.2 m underestimation.

On the East Coast of Scotland, in a southeasterly wind regime, the wave significant height (panel d) is directly linked to wind forcing in the North Sea. Observations shows significant wave heights exceeding 8 m, whereas the coupled ensemble mean peaks at around 6-7 m. Nevertheless, the timing of the wave peak is well captured. Figure S6 confirms this wave underestimation also against Central North Sea oil rig observations. Figure S5 indicates that the strongest winds (in the range of 20-25 $m s^{-1}$) off the coasts of Denmark are underestimated by 3-5 $m s^{-1}$ in all model members.

The highlighted red areas in Figure 4 represent times when elevated river discharge, high tide, strong waves, and storm surge occurred simultaneously. It is this temporal overlap; that produced the severe multi-hazard coastal impacts of Storm Babet. The coupled ensemble is more successful at capturing multi-hazard on the afternoon of 20/10 than in the morning, given its delay in river peak.





## 3.3 Air mass trajectories and moisture origin

To explain the origins of the multi-hazard in storm Babet, we performed air mass Lagrangian trajectories (Section 2.2) using the Met Office global deterministic forecasts to provide insight into the pathways and moisture history of air parcels feeding precipitation during Storm Babet. Backward trajectory analyses were conducted for two trajectory release times on 19 October at 04UTC (Figure 5) and at 22UTC (Figure 6), to trace the origin and thermodynamic evolution of air masses reaching Eastern Scotland. Air parcels were released from near-surface levels (950-970 hPa) over the target region (56.0-57.0°N, 1.0°W-0.0°E),

with a spatial resolution of 0.2° in both latitude and longitude. Each parcel was tracked 16 hours backward and 2 hours forward in time, providing a continuous reconstruction of their pathways and thermodynamic history.

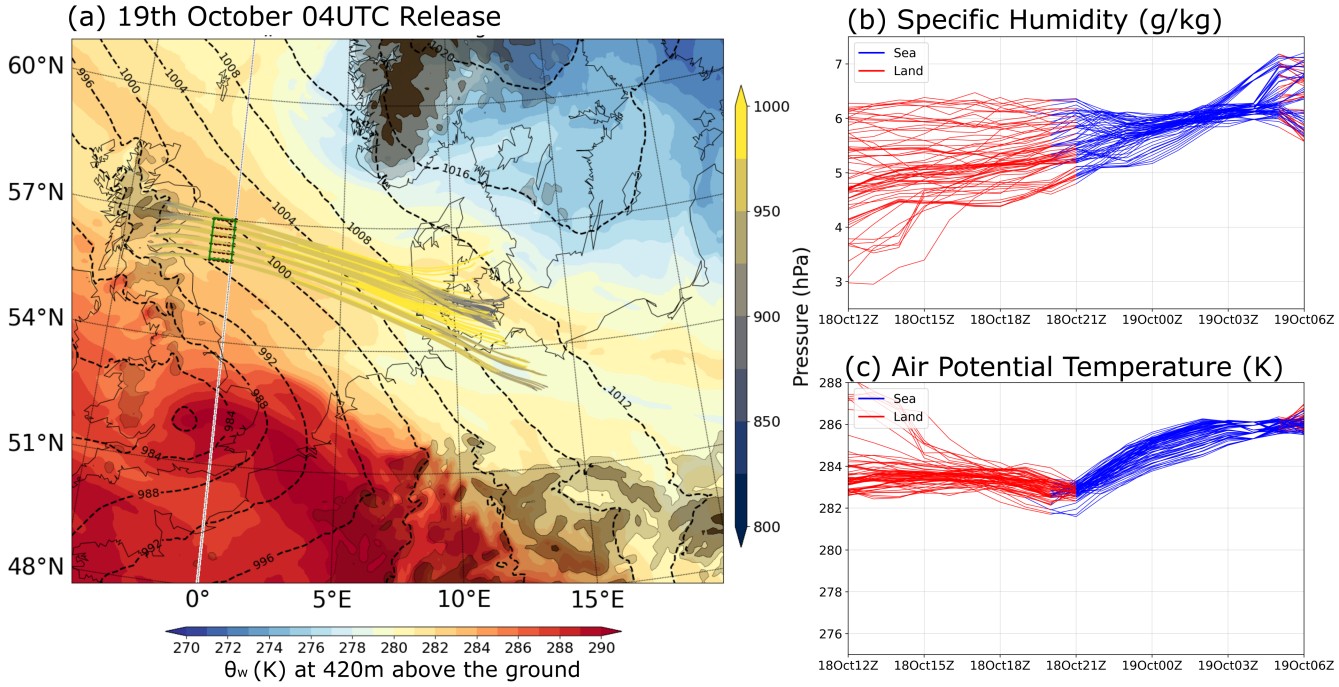

**Figure 5.** Air parcel trajectories and thermodynamic evolution for the 19 October 2023 04UTC release. (a) Wet-bulb potential temperature ($\theta_w$) at 420 m above ground level, overlaid with sea-level pressure (hPa) valid on 18 October 21UTC (midpoint of the trajectory) and 18-hour trajectories (16-hour backward + 2-hour forward) released from 56.0°N-57.0°N, 1.0°W-0.0°E at 925-970 hPa. (b) Specific humidity (g/kg) along the trajectories. (c) Air potential temperature (K). Red: Parcels over land; Blue: Parcels over the ocean

At 04UTC on 19 October (Figure 5a), air parcels originate from the southeast and cross Denmark before entering the North Sea. They accelerate over the sea: the longer marine segment is covered in comparable time to the land crossing (Figure 5b,c). The specific-humidity time series shows the parcels are already moist on entering the marine domain. The small air potential

temperature increase (+2.5 K) reflects limited turbulent heat uptake during the sea crossing, due to limited air/sea temperature difference (Figure S3-4).




By 22UTC on 19 October (Figure 6), the air parcels originated from the east, from cooler and drier land areas before arriving over the North Sea. Initially, these parcels contained lower moisture content, but as they passed over the MHW region, the specific humidity roughly doubles over the sea and potential temperature rises by 5-6 K, indicating respectively intense latent heat and sensible heat flux. This behaviour points to strong air-sea interaction later on 19 October: the North Sea provided an abundant moisture source that primed the atmosphere for heavy rainfall.

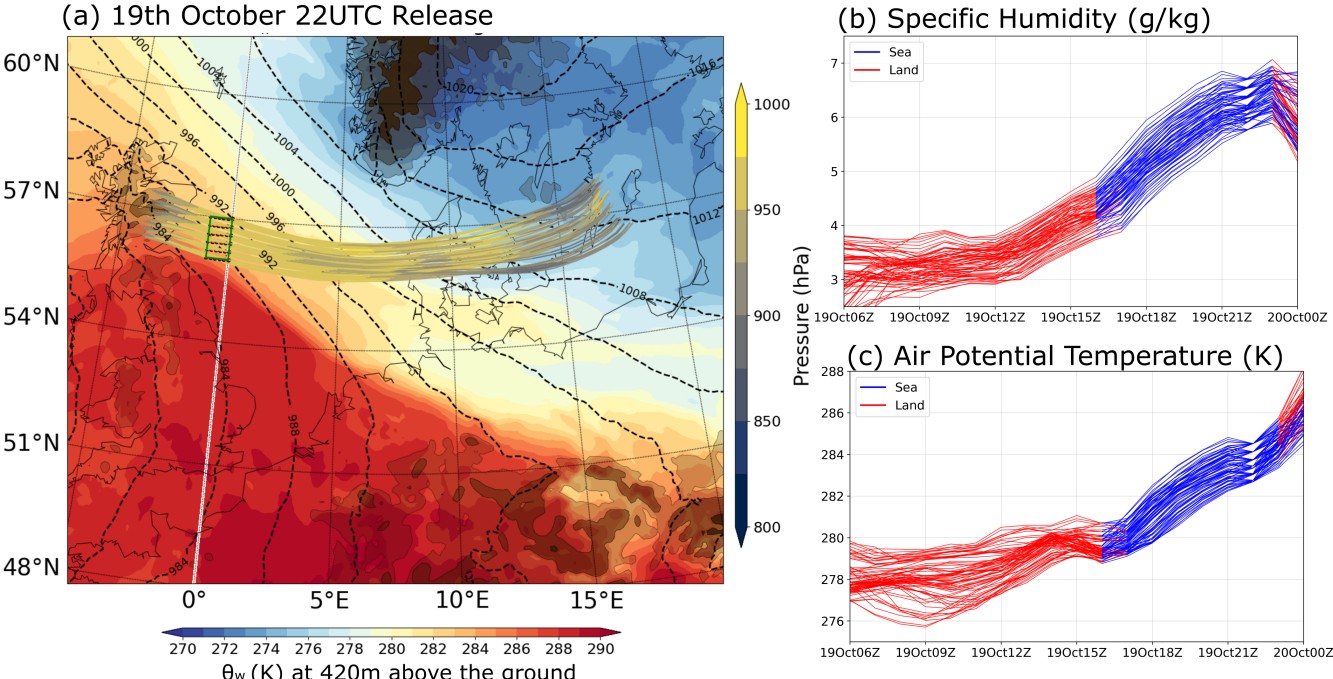

**Figure 6.** Same as Figure 5, but for trajectories released at 19 October 2023 22 UTC. In panel (a), $\theta_w$ (K) is shown at 420 m above ground level and the sea-level pressure (hPa) field is valid at 19 October 2023 15 UTC.

These trajectory analyses show distinct pathways and thermodynamic histories, pointing to the North Sea as a source of moisture for precipitation in Eastern Scotland on the evening of 19 October and on 20 October. Section 4 examines how the MHW preconditioning led to precipitation extremes and multi-hazard amplification.

# 4 Multi-Hazard Amplification by the Marine Heatwave

Following the analysis of the MHW and of storm Babet in Section 3, this section quantifies the direct impact of the MHW, demonstrating how the anomalously warm North Sea served as an amplification factor for an extreme multi-hazard storm event. In this section we first examine the storm's atmospheric intensification (4.1), followed by an assessment of the resulting coastal and hydrological hazards (4.2).





## 4.1 Atmospheric Response and Storm Intensification

To assess the atmospheric response to the MHW, a counterfactual analysis was performed between two 18-member ensemble simulations over the 5-day forecast window from 17-21 October. As detailed in Section 2.4, these experiments represent the observed 2023 event with the marine heatwave (MHW) and a counterfactual scenario where the observed 2023 event occurred over near-climatological sea conditions from 2020 (no_MHW). This approach isolates the impact of the anomalously warm Northwest European Shelf on the storm's development. Figure 7 presents the temporal evolution of key meteorological variables, and clearly shows that the presence of the MHW led to a systematic intensification of the storm system.

The MHW simulation maintained consistently higher SST throughout the 17-22 October period (Figure 7a), by about 0.8-1 °C, the this anomaly was spread throughout the 40-50 m of the Central North Sea.

A consistent atmospheric response to this anomalous warmth is evident in mean sea-level pressure (MSLP) (panel b). The MHW ensemble shows a slightly deeper low-pressure centre, with reductions of 1-2 hPa around 19 October. Although small, this difference is statistically significant and reflects a modest but systematic enhancement of storm intensity. These results align with the synoptic-scale pressure patterns discussed earlier (Section 3.1), where a blocking anticyclone over Scandinavia confined Babet over the North Sea and allowed the low-pressure system to intensify locally. This intensification is likely linked with increased latent heat release from increased precipitation (panel f).

The MHW run consistently produced stronger winds, a small 1% increase especially during the wind decay phase. The model evaluation (Figure S5) showed a realistic temporal evolution of winds, supporting confidence in these results. The trajectory analyses (Section 3.2) and the model evaluation of sea and air temperature (Figures S3 and S4 respectively), showed the air parcels over the North Sea were always cooler than the sea, and in particular on 20 October, meaning the boundary layer was both thermally and (given high wind speed) dynamically unstable. In these conditions, convective downdrafts bring free-tropospheric momentum near the surface, enhancing wind speed. A warmer sea in the MHW experiment resulted in a higher boundary layer (panel e): an increase of 30-50 m relative to the no_MHW case, which means more momentum likely to be carried to the surface. Storm MSLP deepening may also have contributed to stronger wind speeds.

The most direct atmospheric signal of the MHW is seen in latent heat flux (panel d). Fluxes in the MHW simulation were 15-20% higher than in the no_MHW case, especially during 20 October when the storm reached peak intensity. The ensemble mean shows a statistically significant enhancement across much of the storm's evolution. This amplification reflects the greater air-sea humidity contrast provided by the warm MHW waters, but its strength depended strongly on air-mass origin. As shown in the trajectory analysis (Section 3.1), early on 19 October the air parcels were already moisture-laden, winds were weaker, and little additional moisture was taken up. By contrast, later on 19 October the air arrived drier and under stronger winds, enabling vigorous evaporation from the MHW region. This led to a sharp increase in latent heat flux and set the stage for enhanced precipitation over Eastern Scotland.

Panel f shows the accumulated precipitation in Eastern Scotland. Both ensembles reproduce the extreme rainfall totals associated with Babet, but the MHW run yields a consistent enhancement of 5%. While small relative to the storm total, the increase is statistically significant and occurs mainly on the afternoon of 19 October and on 20 October, coinciding with the





**Figure 7.** Time series comparing the MHW (red) and no_MHW (blue) 18-member ensemble simulations over the 5-day forecast period from 17-21 October. For each panel, solid lines represent the mean, dotted lines show the full ensemble range (minimum and maximum), and the shaded areas denote ±1 standard deviation. The panels display: (a) sea surface temperature (K), (b) mean sea level pressure (hPa), (c) 10-metre wind speed (m/s), (d) surface latent heat flux (W/m$^2$), and (e) planetary boundary layer height (m), with each variable averaged over the Central North Sea. Panel (f) shows the accumulated precipitation over Eastern Scotland (red box in Figure 3, mm, left axis) and the difference between the ensemble means (MHW - no_MHW, black line, right axis). Red asterisks indicate statistical significance at the 99 % confidence level (p < 0.01) based on a paired t-test between corresponding ensemble members ($\mathrm{MHW}_i - \mathrm{no\_MHW}_i$).





period when dry air parcels crossed the warm North Sea and acquired substantial additional moisture. In contrast, the MHW
exerted little influence on 19 October, when incoming air masses were already humid. Importantly, the timing of the enhance-
ment corresponds to the period of strongest latent heat flux and boundary layer deepening, reinforcing the link between the
ocean anomaly and atmospheric intensification. The magnitude of the increase also aligns with Clausius-Clapeyron expecta-
tions for the observed SST anomaly (7% more moisture per degree of warming), but crucially its manifestation depended on
the interplay between the marine heatwave and air-mass origin.

## 4.2   Coastal and Hydrological Hazards

The atmospheric intensification linked to the North Sea MHW extended into the hydrological and coastal hazards, amplifying
river discharge, surge, and wave power. To quantify these impacts, we analyse the same pair of 18-member ensemble forecasts
described in Section 2.4, spanning 17-21 October. Figure 8 presents the temporal evolution of these hazard indicators: river
discharge at river mouths, wave power and surge averaged over the red box of Figure 3 highlighting the role of anomalously
warm SSTs in amplifying multi-hazard risk.

Figure 8a,c show that both the river Tay and the South Esk experienced a systematic increase in flow in the MHW ensemble.
By 20-21 October, flows were enhanced by around 12-18% ($\sim$30 m$^3$ s$^{-1}$). The South-Esk peaked earlier, on 20 October,
whereas the Tay catchment peaked later, on 21 October, reflecting different basin response times. Importantly, both rivers
burst their banks during Storm-Babet, and the model indicates that the MHW amplified the peak flows, thereby raising the
probability of fluvial flooding.

The coastal storm surge Figure 8(b) was amplified by about 20% in the MHW ensemble on 19 October ($\sim$3 cm averaged
on the Eastern Scotland box), consistent with slightly stronger near-surface winds (+1%) and a deeper storm core (widespread
$\sim$2 hPa mean sea-level pressure anomaly) on this day. The persistence of the anomaly nonetheless illustrates how MHW-driven
intensification can increase the frequency of hazardous water levels, even when tidal phasing mitigates the outcome.

Finally, Figure 8(d) shows that wave power, defined as the energy flux carried by waves (proportional to wave height squared
times wave period), was elevated in the MHW case by about 9% on 19-20 October. This enhancement is consistent with the
$\sim$1% increase in wind speeds, arising from greater boundary-layer depth and enhanced thermal instability over warmer waters.
Indeed, waves are an integrator of wind speed, and a 1% increase in wind speed in our case leads to 3% wave height increase,
and therefore 9% wave power increase. Stronger and more energetic waves increase the risk of coastal erosion and overtopping
of sea defences.

At the end of 19 October, the MHW run shows that wave power was increasing while the surge had already started to go
down. Even though the surge was smaller than earlier, it was still higher than in the no_MHW case. This means that the peaks
of surge and waves did not fully happen at the same time, but the MHW still made both hazards larger than they would have
been otherwise. If these waves happen at the same time as high water levels, they can also make it harder for rivers to empty
into the sea, which can worsen flooding in estuaries. Our model does not link the rivers directly with the sea, but this kind of
interaction is well known, and it shows why it is important to look at all the hazards together.




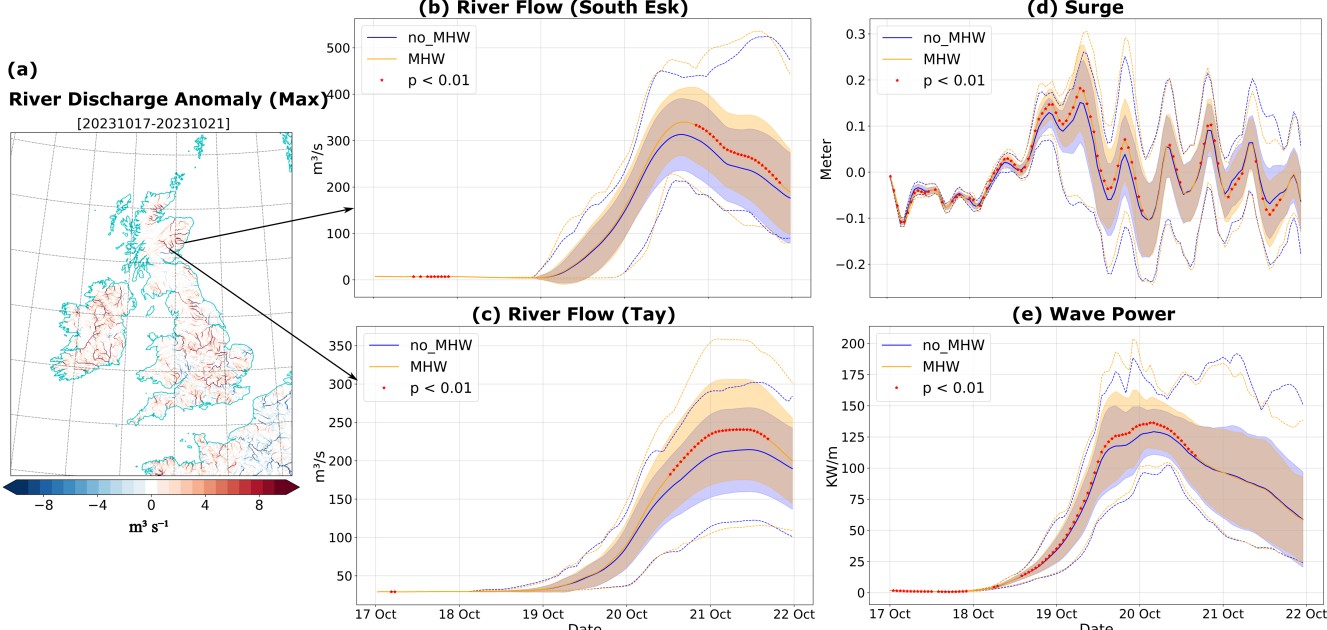

**Figure 8.** (a) River discharge anomaly (2023 - 2020) b) river flow in the mouth of river South Esk (c) river flow in River Tay, d) surge on the Eastern Scotland red box (sea only), surge is computed using ocean model total water level - tide-only run, e) Wave power in Eastern Scotland red box (sea only). Red asterisks indicate statistical significance at the 99 % confidence level ($p < 0.01$) based on a paired t-test between corresponding ensemble members ($\mathrm{MHW}_i - \mathrm{no\_MHW}_i$).

These diagnostics demonstrate how modest SST anomalies can propagate through multiple pathways to amplify storm impacts. Even a small increase in wind, precipitation and decrease in pressure can integrate over time and space and generate much larger river, surge and wave power increases. The MHW systematically raised all components, thereby increasing the overall hazard envelope and amplifying the multi-hazard events (river, surge, tide, wave) shown in the observations on 20 October (Figure 4). This highlights the potential for marine heatwaves to act as multi-hazard amplifiers: even relatively small SST anomalies (∼1°C) can trigger a chain of processes that lead to greater and longer-lasting riverine, coastal, and wave impacts.

# 5 Discussion

## 5.1 Limitations of the study

This study has several limitations that should be considered when interpreting the results. The analysis focuses on a single case study, storm Babet, meaning that the identified processes cannot yet be assumed to apply to all extratropical cyclones. Every storm has unique synoptic characteristics, so generalisation requires a larger event set. In addition, the experimental design




relies on the Met Office regional coupled prediction system which, although state-of-the-art for UK forecasting, represents
only one model configuration and is subject to biases. The difference between 2023 and 2020 sea states used in the coupled
model tends to underestimate the MHW amplitude relative to observations (Figure 1), implying that the simulated anomaly
represents a conservative estimate of the true MHW signal (it corresponds to an anomaly with a 2003-2022 baseline, rather than
1982-2012). Indeed, repeating this study with an atmosphere-only ensemble prescribing the observed MHW anomaly showed
a 6-7% increase in precipitation, slightly higher than the 5% of the coupled system (not shown). Although the coupled system
remarkably captured individual hazards, timing of increase, it did underestimate most of them, so it is possible that the MHW
amplification of these hazards may also be underestimated. Nevertheless, the coupled system captured very well the sea surface
temperature cooling (Figure S3), giving confidence in how much energy was extracted from the North Sea by the storm in the
coupled forecast. Furthermore, the regional model setup focuses on the Northwest European shelf and surrounding regions, so
feedbacks on larger-scale circulation features influencing the storm may not be fully represented.

## 5.2    Implications for predictability and risk

Despite these limitations, the results highlight important implications for understanding and predicting the role of MHWs for
mid-latitude storm impacts. The results show that MHWs can intensify multiple hazards associated with extratropical cyclones.
In this case, the warm anomaly in the North Sea enhanced latent heat fluxes, which in turn supported greater atmospheric
instability and additional precipitation over Eastern Scotland. However, the expected thermodynamic effects alone do not
explain the response; the amplification also depended on a shift in the air-mass origins, showing that local and large-scale
factors acted together. The model experiments also showed consistent increases in river discharge, storm surge, and wave
power under MHW conditions, suggesting that the influence of the ocean surface extended across several components of the
hazard chain. These findings highlight that even a modest SST anomaly was sufficient to produce systematic differences,
underlining the sensitivity of shallow seas to air-sea coupling.

We can formalise these pathways, in a causal network that links anomalous ocean heat content to surface fluxes, storm
intensity, and downstream hazards (Figure 9). This framework provides a structure for attributing observed responses and
testing whether similar connections appear in other cases. With a larger event set, the approach could establish more robust
relationships between North Sea heat content and storm severity, allowing generalisation beyond single events.

These findings have broader implications for both predictability and risk assessment. Marine heat content is an observable
quantity that can be monitored and predicted at sub-seasonal to seasonal(S2S) timescales. If storms with similar characteristics
to Babet occur during periods of elevated heat content in the North Sea, they are likely to be amplified in a comparable way. This
provides a physical basis for using ocean heat content as a precursor variable in forecasts of storm impacts. It also shows that
the interaction between MHWs and storms is not a rare coincidence, but rather a systematic process that should be integrated
into risk management frameworks. Storms tend to always end MHWs, and this example shows that MHWs can amplify the
storms that end them.




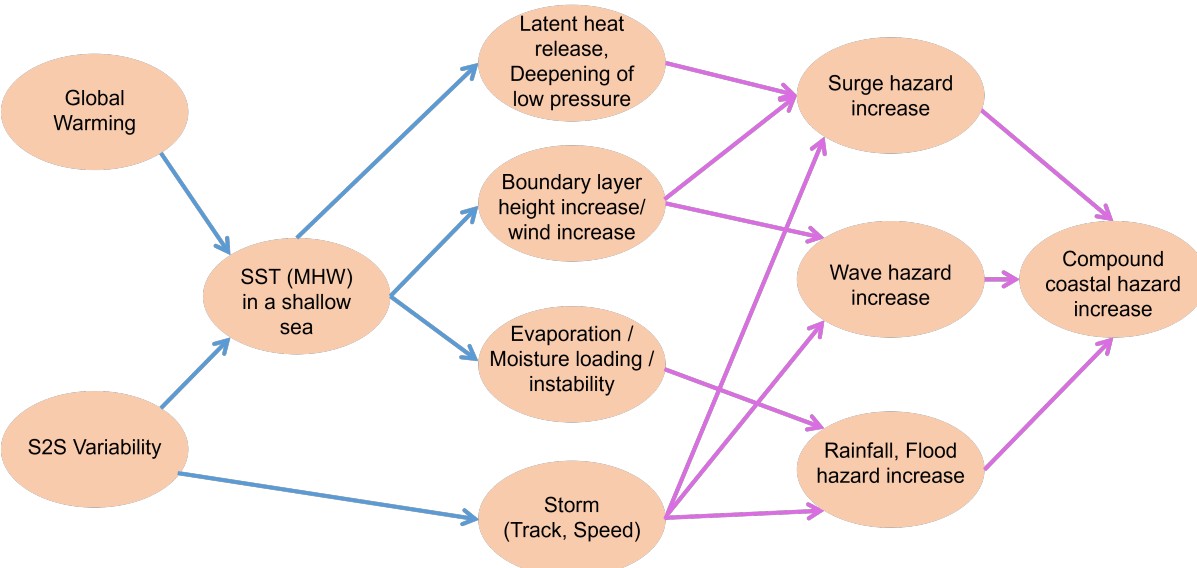

**Figure 9.** Causal network showing physical linkages between MHWs in shallow seas and multiple hazard components. Arrows represent proposed causal relationships informed by the Storm Babet case study. Blue arrows denote process links up to the storm and atmospheric responses. Magenta arrows denote integrating responses where small atmospheric changes amplify when accumulated over space and time.

## 5.3 Implications for climate projections

The MHW that coincided with Storm Babet represents both internal variability and the influence of long-term climate change. The observed North Sea anomaly can be interpreted as an analogue for future average conditions in a warmer world: indeed, the North Sea warms faster than the global warming trend, and warmed by 1°C between 2003-2022 and 1982-2002 (Figure 1), and 2023 was 0.9°C warmer than 2003-2022, which, if the past trend continues, would be close to a 2023-2042 average. Increases of up to 4.1°C for the Central North Sea and 4.5°C for the Southern North Sea are possible for the last two decades of the century compared to the first two (Tinker et al., 2024). From this perspective, Babet is not only an example of present-day hazard amplification, but also a prototype of how storms may evolve as mean ocean temperatures rise. When such warming becomes the baseline rather than an extreme, the amplification observed here would apply to a much larger set of storms.

Nevertheless, to establish whether these mechanisms are general or case-specific, it is important to extend the analysis beyond Babet. Different storms are likely to show distinct balances of response; for instance, larger changes in mean sea-level pressure but smaller changes in precipitation. Analysing a broader set of events will make it possible to test whether the causal pathways identified here recur across storms, or whether they depend on particular synoptic configurations. This comparative approach will provide a more robust basis for linking marine heatwave conditions to storm impacts.





## 6 Conclusions

This study examined how the prolonged autumn 2023 marine heatwave (MHW) in the North Sea affected the evolution and multi-hazard impacts of the storm which ended the MHW: storm Babet. Using a high-resolution regional coupled atmosphere-ocean-wave model (UKC4), we ran ensemble forecasts comparing the observed event (MHW) with a counterfactual simulation initialised with ocean conditions representative of last 20-year climatology (no_MHW).

The pre-existing MHW was generated by persistent anticyclonic conditions a month and a half before the storm happened, and acted as a local amplifier of storm hazards. Indeed, the North Sea was anomalously warm by 0.8-1°C throughout its shallow depth, meaning no cold water could be entrained from deeper layers like in deeper parts of the ocean. The long-lasting (>48 h) cold and dry low-level jet generated by the storm over the North Sea was energised by the MHW which raised surface latent heat flux by about 15-20% and increased atmosphere instability, increasing wind speed by 1%. This produced a modest but statistically significant 5% increase in rainfall over eastern Scotland and 1-2 hPa deepening of the cyclone. This atmospheric intensification translated into stronger hydrological and coastal impacts. Higher rainfall increased peak river discharge by 12-18%, stronger winds increased coastal wave power by ∼9%, and stronger winds in combination with cyclone deepening increased the storm surge by ∼20%. The MHW supplied excess heat and moisture that broadened the storm's multi-hazard footprint, especially when drier air masses crossed the basin. Future work should test these mechanisms across more storms and modelling systems to assess generality and sensitivity.

This case identifies a clear physical pathway linking prior increased ocean heat content in a shallow shelf sea to the severity of the impacts of an extratropical cyclone. This finding has implications for operational forecasting, suggesting that regional MHWs can serve as a valuable conditional precursor for S2S predictions of extreme storm impacts.

This event is a prototype for storms in a warmer climate: the warming level of the North Sea SST at the time of storm Babet was representative of a 2023-2042 average, if warming from the last 40 years is extrapolated. As anthropogenic warming makes the North Sea MHWs more frequent and intense (Cornes et al., 2023b; Tinker et al., 2024), the compound amplification of rainfall, surge, and waves seen during Storm Babet may become systematic for storms with similar characteristics to storm Babet.

This argues for integrating coupled ocean-atmosphere processes into adaptation planning and multi-hazard risk assessments. Indeed, the Met Office and Environmental Agency operational forecasting system at the time of storm Babet used inconsistent forcing between atmospheric, river, ocean/wave and surge systems, and each hazard was analysed independently. A coupled system with good skills across multiple earth system components, as illustrated is this paper, would enable a consistent approach to multi-hazard forecasting, and a whole-system understanding of storm hazards, also including impacts on marine ecosystems (Berthou et al., 2025b; Partridge et al., 2025).

*Code and data availability.* Glider data are available from the BODC Deployment Catalogue at https://platforms.bodc.ac.uk/deployment-catalogue/, observed river data is available from https://timeseriesdoc.sepa.org.uk/, observed wave data from https://wavenet.cefas.co.uk/ and observed



surge data from https://ntslf.org/. Coupled model output is available in https://doi.org/10.5281/zenodo.17609383. Code used to generate the figures of this article is available from https://doi.org/10.5281/zenodo.17609383.

*Author contributions.*  PG: Conceptualization, Methodology, Formal analysis, Investigation, Data curation, Visualization, Writing – Original
Draft, Writing – Review & Editing. SB: Conceptualization, Methodology, Formal analysis, Investigation, Data curation, Visualization, Supervision, Writing – Original Draft, Writing – Review & Editing. TGS: Conceptualization, Methodology, Investigation, Supervision, Writing – Review & Editing. AV: Methodology, Software, Investigation, Writing – Review & Editing. SM: Software, Validation, Resources, Writing – Review & Editing. JMC: Software, Resources, Writing – Review & Editing. ACP: Resources, Validation, Writing – Review & Editing. YJP: Resources, Writing – Review & Editing. RR: Supervision, Writing – Review & Editing. MAB: Supervision, Writing – Review & Editing.

*Competing interests.*  The authors have no competing interests.

*Acknowledgements.*  PG was supported by the AFESP research fund at the University of Reading. SB was supported by the Met Office Hadley Centre Climate Programme funded by DSIT. We thank the Met Office for access to the Regional Coupled System (UKC4) through the MONSooN2 platform, and JASMIN for data processing and storage. Glider data are funded by the Met Office Supercomputer Observations Programme.



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
