# Peer review of "Marine heatwave amplifies extreme multi-hazards of extratropical cyclone Babet"

_EGUsphere, 2025_

## Referee Comment (RC1)

Marine heatwave amplifies extreme multi-hazards of extratropical cyclone Babet

Review for NHESS

Goswami et al present their case study of the interaction between the autumn 2023 marine heatwave (MHW) and extratropical cyclone Babet, demonstrating through the use of a counterfactual scenario that the strong MHW influenced the development and hazards of the storm. While certain elements could use slightly more explanation as to the reasoning behind choosing that method, it is based securely in peer-reviewed methods which are combined and applied to a novel case; this is a well-structured, clearly written paper that after some minor revisions would be an excellent candidate for publishing. Please see my comments below for the specifics.

**Several points require a little more explanation:**

Line 153-154: Why is the atmosphere ensemble based and the wave components deterministic?

Line 300-301: Can you add a justification for why you choose the two times you did for the lagrangian parcel tracking?

Page 12 paragraph 2 : You mention that the peak wind (and thus the surge amplitude) is underestimated by the models, but barely reference this in the discussion. Do you have an idea as to why this is underestimated?

**Other:**

Line 74: I think there's a word missing "the use of the global model enables *us* to..."

L162-164: Mentioning that the 2020 run is started 3 years and 2 days before the forecast model could lead to confusion – I understood this initially as the no_MHW ocean being allowed to run for 3 years before the simulation starts, which would introduce great amounts of uncertainty. Rephrasing for clarity is suggested.

L194: Consider specifying that this is the specific heat capacity of sea water and not just water in general.

Section 3.1, paragraph 2: The description of weather regimes/how to read Fig 2a comes after conclusion drawn from 2a, and thus the 1/3/6/12/16 mentioned have little significance for the reader not familiar with these weather regimes. I would move the lines "Weather regimes between 1-10...autumn and spring." to after the first sentence in the paragraph.

Line 209: typo – *anticyclonic*

Section 3.2 paragraph 1: I think an extra reference here would be useful, as the only reference is Kendon et al at the beginning of the paragraph.

Line 291: typo – observations *shows*

Line 296: extraneous semicolon

Line 330: maybe specify Northwestern European shelf *sea*? I wondered why the shelf itself was warm

Figure 7 – MHW is yellow but the caption says red, also the solid MHW line is really difficult to see. I suggest trying to find a better way of indicating significance than the red asterisks (shading of the background perhaps?)

Figure 8 – It may be useful to add the red box of the domain to this map as well, as you refer to it twice in the caption alone

One last suggestion I have is to add a figure to help understand the evolution/life/path of Babet. I had trouble picturing the path of the storm or how East Scotland was the area hardest hit with rain despite it coming from the south. I understand it is difficult what with Babet being composed of the multiple different lows combining, but if possible (and useful) I think this could add clarity. Perhaps through the use of (radar) images as in Kendon et al(2023)?